# Q-Function-Based Traffic- and Thermal-Aware Adaptive Routing for 3D Network-on-Chip

**Seung Chan Lee** [1] and **Tae Hee Han** [2],*

[1]    Department of Semiconductor and Display Engineering, College of Information and Communication
     Engineering, Sungkyunkwan University, Suwon, Gyeonggi-do 16410, Korea; chan0614@skku.edu
[2]    Department of Artificial Intelligence, Sungkyunkwan University, Suwon, Gyeonggi-do 16410, Korea
*    Correspondence: than@skku.edu; Tel.: +82-31-299-4587

**Abstract:** Die-stacking technology is expanding the space diversity of on-chip communications by leveraging through-silicon-via (TSV) integration and wafer bonding. The 3D network-on-chip (NoC), a combination of die-stacking technology and systematic on-chip communication infrastructure, suffers from increased thermal density and unbalanced heat dissipation across multi-stacked layers, significantly affecting chip performance and reliability. Recent studies have focused on runtime thermal management (RTM) techniques for improving the heat distribution balance, but performance degradations, owing to RTM mechanisms and unbalanced inter-layer traffic distributions, remain unresolved. In this study, we present a Q-function-based traffic- and thermal-aware adaptive routing algorithm, utilizing a reinforcement machine learning technique that gradually incorporates updated information into an RTM-based 3D NoC routing path. The proposed algorithm initially collects deadlock-free directions, based on the RTM and topology information. Subsequently, Q-learning-based decision making (through the learning of regional traffic information) is deployed for performance improvement with more balanced inter-layer traffic. The simulation results show that the proposed routing algorithm can improve throughput by 14.0%–28.2%, with a 24.9% more balanced inter-layer traffic load and a 30.6% more distributed inter-layer thermal dissipation on average, compared with those obtained in previous studies of a 3D NoC with an 8 × 8 × 4 mesh topology.

**Keywords:** die-stacking; 3D network-on-chip; heat dissipation; runtime thermal management; Q-function; reinforcement machine learning; Q-learning

---

## 1. Introduction

Since the mid-2000s, a chip multiprocessor (CMP) has been widely used to overcome the limitations concerning instruction-level parallelism and power walls in a single-thread/core processor [1]. However, the ever-increasing traffic between the processing elements created bottlenecks in conventional bus-based CMPs [2]. Initially, a 2D network-on-chip (NoC) was proposed for mitigating the complexities in the on-chip interconnection network [3]. Although a 2D NoC has the advantages of high scalability and simple fabrication structure, high performance is not guaranteed, owing to the rapid deterioration in packet latency associated with increasing physical distances as the number of processing cores increases [4]. With the advent of through-silicon-via (TSV)-based 3D integrated circuits, structural changes have entered a new phase, beyond 2D NoCs [5]. A 3D NoC-based CMP architecture that fully exploits die stacking using TSV technology offers a wider bandwidth, a lower packet transfer delay and a smaller layout footprint with shorter average internode distances than a conventional 2D NoC [6].

However, the increased thermal density in a 3D NoC eventually leads to saturation in performance and reliability, requiring additional cooling circuitry, such as a heat sink [7]. The stacked structure of the 3D NoC results in a longer heat dissipation path and a different inter-layer cooling efficiency [8],

as shown in Figure 1. The processor nodes in the top layer farthest from the heat sink encounter severe thermal problems, intensifying the heat imbalance across the layers of the 3D architecture [9]. The importance of thermal management algorithms is discussed not only in 3D NoC, but also in microfluidic processes that are widely used in other applications, such as fluid networks or chips [10]. Moreover, the TSVs used in stacked structures are constrained by larger bonding areas, complicated scaling processes with smaller feature sizes and sharp decreases in the yield as the number of TSVs increases [11,12], as shown in Figure 2. The severe drop in yield causes a transition to a non-stationary irregular (NSI) mesh, rather than the fully connected mesh of a 3D NoC topology, thereby reducing the packet routing flexibility [13]. This leads to decreased router utilization efficiency, which, in turn, increases the risk of deflection of the traffic load distribution, further exacerbating the thermal problems [14].

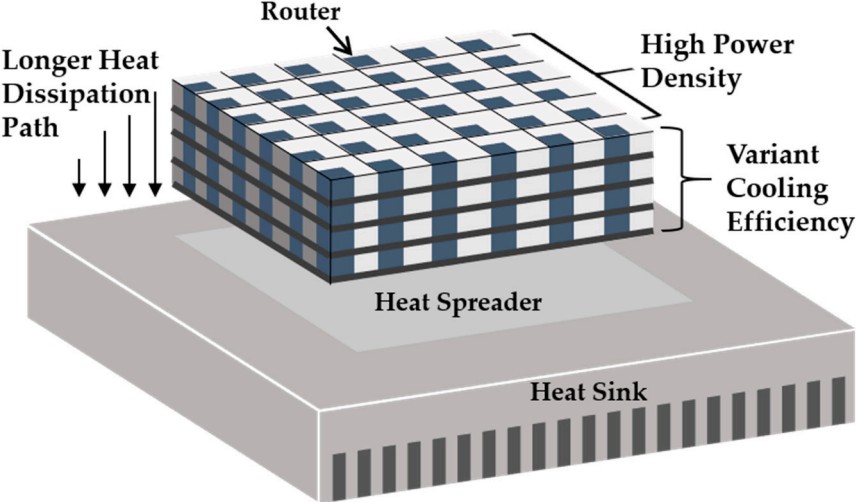

**Figure 1.** Thermal considerations in 3D stacked network-on-chip.

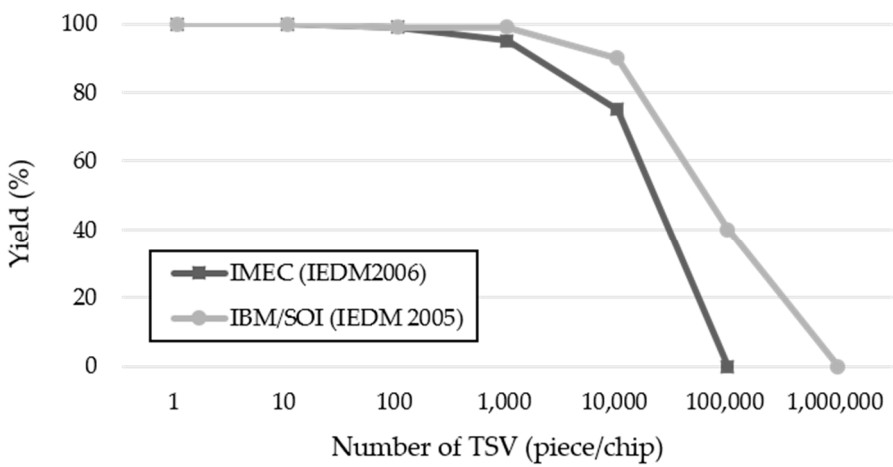

**Figure 2.** Dependence of yield on the number of through-silicon-via (TSV) integrations for different manufacturing processes [12].

For temperature management in a 3D NoC, numerous runtime thermal management (RTM) methods have been proposed. RTM is classified into proactive pre-operation approaches and reactive post-operation approaches, as presented in Table 1. Proactive RTM collects thermal information to predict the operating duration of each node before reaching a throttling state and the thermal distribution of the 3D NoC is adjusted by routing to nodes with larger thermal margins [9,15,16]. Proactive RTM is advantageous in preventing the occurrence of throttled nodes. However, proactive

RTM algorithms route packets with information predicted at a specific time and it is challenging to address temperature changes caused by sudden traffic changes. Therefore, proactive RTM cannot guarantee that router temperatures will remain stable for normal operation. Reactive RTM in 3D NoCs manages temperature using a post-operation, such as forcing a node with an alarm state to become idle [5,6,17]. Reactive RTM algorithms generate routing policies, mainly focusing on traffic congestion as they cannot estimate the thermal margin. However, excessive blocking occurs as the number of throttling nodes increases, resulting in a negative performance, owing to reduced routing flexibility.

**Table 1.** Comparison between proactive and reactive runtime thermal management in 3D network-on-chips (NoCs).

| | Proactive RTM | Reactive RTM |
|---|---|---|
| **Method** | Predictive manner, precaution | Post-operation, blocking |
| **Pros** | Preventing throttled nodes in advance | Concentrating on performance without thermal pre-consideration |
| **Cons** | Computation overhead for prediction | Cannot prevent throttled nodes in advance |
| **Related works** | PTDBA [9], Cool-elevator [15], PTB3R [16] | TLAR [5], TAAR [17], TTABR [6] |
| **Detailed operation** [18] | 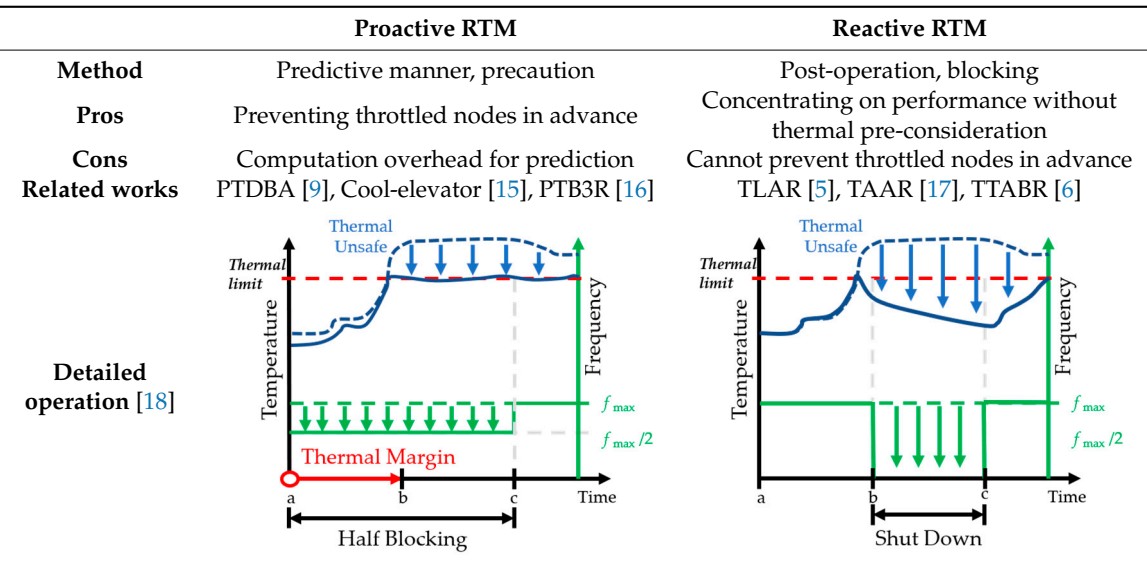 | |

Considering the factors affecting the performance and reliability of 3D NoCs, it is not always possible to simultaneously achieve the required performance and thermal management through a heuristic approach [7]. Moreover, manually designing algorithm rules and strategies demands substantial engineering efforts. As a result, recent studies have proposed routing algorithms based on reinforcement machine learning, in order to automatically generate control policies for optimal results [19]. As reinforcement learning requires no human engineering or labeled training data when creating policies, it has the strength to directly generate optimal decision policies by learning runtime NoC states [20].

In adaptive routing algorithms, the Q-routing method, which combines reinforcement machine learning and routing strategy, has been proposed [21–23]. In conventional Q-routing, each node learns the network congestion status on the basis of local and global information, i.e., packet transfer delay, and stores the learned outcomes in a Q-table as scores that can determine an optimal packet output channel [21]. Stored scores (Q-values) are used as a criterion for determining the priorities of all paths. The Q-function gradually reflects traffic load information by updating the Q-value with a weighting of the stored scores in the Q-table to produce a new estimate from the existing neighboring nodes [22]. A routing scheme adopting this learning has a strong potential for significantly enhanced performance compared with conventional thermal-aware routing methods, by adapting the paths of packets to time-varying traffic loads in 3D NoCs [23]. Furthermore, the near-optimal solution of Q-learning is advantageous for constructing an optimal routing policy that considers performance and thermal management comprehensively in a 3D NoC, while considering more diverse factors than a 2D NoC [7]. However, the Q-table size overhead is problematic, as decisions are made using information regarding all paths for a specific node [23].

The contribution of this study is the proposal of a Q-function-based traffic- and thermal-aware adaptive routing (QTTAR). QTTAR is a Q-learning-based adaptive 3D routing algorithm for improving overall node utilization by balancing the distribution of inter-layer traffic and providing a more accurate congestion analysis to mitigate performance degradations due to RTM. Furthermore, the proposed algorithm mitigates differences in inter-layer cooling efficiency by balancing the distribution of overheated regions by layer. QTTAR recognizes regional congestion and hotspots by learning rapidly changing networks. Based on the Q-table, QTTAR generates an optimal policy and creates a routing decision. We describe the proposed routing algorithm in the context of two purposes: obtaining a high level of routable path diversity and developing a Q-function-based routable direction selection strategy. The process of collecting routable paths involves an adaptive intra-layer scheme and a downward interlayer scheme, with the final direction selection process determined based on the Q-value stored in the Q-table.

The rest of this paper is organized as follows. In Section 2, we introduce the Q-learning-based algorithm and other traffic- and thermal-aware routing algorithms for a 3D NoC. Section 3 describes the proposed QTTAR algorithm. In Section 4, we discuss the simulation results. Section 5 presents conclusions of the study.

## 2. Related Work

### 2.1. Routing Algorithms Using Runtime Thermal Management (RTM) in a 3D Network-on-Chip (NoC)

A number of traffic- and thermal-aware adaptive routing algorithms have been proposed to solve the temperature and traffic congestion problems in a 3D NoC. They mainly differ in the routable direction selection strategy and the RTM method. The neighbors-on-path and regional congestion awareness strategies are primarily used for routing direction selection [24,25]. The routable direction selection strategy is an approach for selecting an optimal direction to minimize congested nodes until a packet reaches its destination. According to the RTM method, the routing algorithm of a 3D NoC is classified as either proactive [9,15,16] or reactive [5,6,17].

#### 2.1.1. Routing Algorithms Using Proactive RTM in a 3D NoC

Proactive thermal-budget-based beltway routing (PTB3R) was proposed for identifying potential thermal hotspots based on the remaining active time of nodes before throttling [16]. The PTB3R approach introduced the concept of calculating a mean time to throttle (MTTT) based on current temperature difference and temperature consumption rate in order to to predict the thermal margin of the router before throttling. It attempted to minimize the number of throttling nodes by sending packets to areas with high MTTTs.

Thermal-aware dynamic buffer allocation for proactive routing (PTDBA) balances thermal distribution by adjusting the depth of the router's input buffer [9]. The PTDBA bypasses packets by inducing traffic congestion toward a router with a high rate of temperature rise. Bypassed packets are sent to a non-congested region with a low rate of temperature rise, thereby improving thermal balance and reducing packet congestion.

However, the temperature consumption rates for PTB3R and PTDBA are both calculated from traffic load information at a certain time in the past, which inhibits an accurate reflection of the current state of the network. Moreover, differences between predictions and actual network conditions lead to a failure in preventing throttled nodes, thereby reducing the efficiency of proactive operations.

#### 2.1.2. Routing Algorithms Using Reactive RTM in a 3D NoC

Transport-layer assisted routing (TLAR) balances the traffic load between layers, through separate routing in vertical and horizontal directions, based on topology information [5]. TLAR prefers downward routing and distributed traffic to achieve a thermal balance between layers with lateral routing through the selective use of deterministic and adaptive methods. However, when horizontal

routable directions in the NSI mesh are insufficient, the TLAR performs mainly downward routing. Thus, the traffic load in the bottom layer rapidly increases, reducing the traffic load imbalance between layers.

Topology-aware adaptive routing (TAAR) provides a novel cascaded algorithm that dynamically adjusts the packet routing mode based on network topology [17]. Concurrently, TAAR mitigates traffic imbalances between layers using queuing analysis theory. TAAR transmits packets to a non-congested minimal region, based on regional traffic information at the time of routing. However, contention still occurs, owing to an excessive concentration of packets in the minimal region, which may worsen the thermal imbalance.

A traffic- and thermal-aware adaptive beltway routing (TTABR) was introduced to solve imbalanced traffic and temperature distributions by dynamically selecting non-minimal or minimal routing paths [6]. The addition of non-minimal paths ensures the number of routing paths, reducing excessive contention problems in minimal routing regions. However, as a packet is sent to a bypass path using only traffic information at the routing point, the packet can still reach a hotspot region, as the information to determine actual hotspots can be insufficient.

### 2.2. Q-Learning-Based Routing Algorithm

A Q-learning-based routing scheme alleviates packet transfer delays caused by network congestion in a 3D NoC, with the strength to directly generate optimal decision policies by learning runtime NoC states [7]. A learning packet containing traffic condition information is used to update the Q-table and the optimal output channel is determined based on the stored values. The proposed routing scheme achieves higher throughput and lower power consumption using up-to-date congestion values. However, as the network size increases, the Q-table can cause excessive area overhead because Q-values for all possible route paths are stored. Moreover, existing Q-learning-based routing methods do not incorporate RTM to address thermal management.

The reactive and proactive RTM-based routing techniques described can resolve the overall thermal and contention problems of 3D NoCs, but inter-layer traffic imbalances and performance degradations caused by network congestion still remain challenges. In addition, previous studies using Q-learning did not consider 3D topology and thermal awareness when constructing the Q-table. Therefore, we propose a QTTAR algorithm that adopts RTM and also learns the traffic state of the 3D NoC to enable routing policies to be fully aware of congested regions.

## 3. Q-Function-Based Traffic- and Thermal-Aware Adaptive Routing (QTTAR)

Existing 3D NoC routing algorithms with RTM reduce thermal imbalances. However, traffic congestion owing to frequent blocking from the RTM mechanism and thermal considerations lead to performance degradation. QTTAR adopts the blocking mechanism of reactive RTM and simultaneously aims to alleviate performance degradation and inter-layer thermal imbalance. QTTAR generates routing policies to enable the paths of packets to be fully adapted to the time-varying traffic conditions. To prevent cyclic dependencies and ensure the diversity of routable directions that is important for maximizing the efficiency of the optimal Q-learning solution searching process, the first step of QTTAR involves collecting deadlock-free routing directions based on the topology and location of throttled nodes. In the second step, a policy is created to minimize latency by selecting the final direction via decision-making based on traffic congestion. Through these steps, QTTAR improves the inter-layer traffic load distribution, balances the per-layer biased utilization, performs thermal management and ultimately improves performance. The details of the process are as follows.

**Step 1.** Obtaining a high level of routable path diversity—we divide the routing process into intra- and inter-layers to prevent deadlock generation between the vertical and horizontal directions and to create a candidate group of directions for a packet to proceed, based on information regarding the throttling node;

**Step 2.** Q-function-based routable direction selection strategy—we check the buffer status of each node and score traffic information for each direction. Next, each node updates the Q-table after learning the state of network congestion and, based on the updated information, routes the packet to a non-congested direction among the candidate groups created in the preceding step.

### 3.1. Obtaining a High Level of Routable Path Diversity (Step 1)

The first step in QTTAR is to identify the throttling nodes for a minimal region from the current node to the destination node, then create a candidate group of deadlock-free directions. When the routing function is deterministic, the flexibility of the routable direction decreases rapidly if the minimal region is saturated with throttling nodes. In this case, there are few deadlock-free directions to choose from during the Q-table-based routing decision. Thus, we introduce an adaptive routing function to provide multiple deadlock-free directions. The selection of deadlock-free directions is configured independently using intra- and inter-layer routing to prevent cyclic dependency between the horizontal and vertical directions. Figure 3 shows the flow chart of the first step in QTTAR, where $N_C$ and $N_D$ represent the current node and destination node, respectively. The intra-layer routing contains a 2D NoC deadlock-free "odd–even" routing algorithm [26] providing even path diversity, whereas the inter-layer routing is based on a downward scheme. Existing thermal-aware routing algorithms dominantly use downward schemes despite the fact that routable paths still exist in the XY plane, resulting in an excessive thermal concentration in the lower layers [6,17]. Thus, QTTAR uses odd–even routing when $N_C$ is located above the throttling layer to increase the flexibility of the routing function and maximize the efficiency of Q-table-based decision making. Finding a route adaptively increases the diversity of paths compared with a deterministic approach. However, the computational complexity is $O(4^N)$ when all paths are found adaptively for the intra-layer region, which makes the overhead of the selection process very high [5]. The computational overhead is reduced by only searching for throttling nodes for minimal path regions.

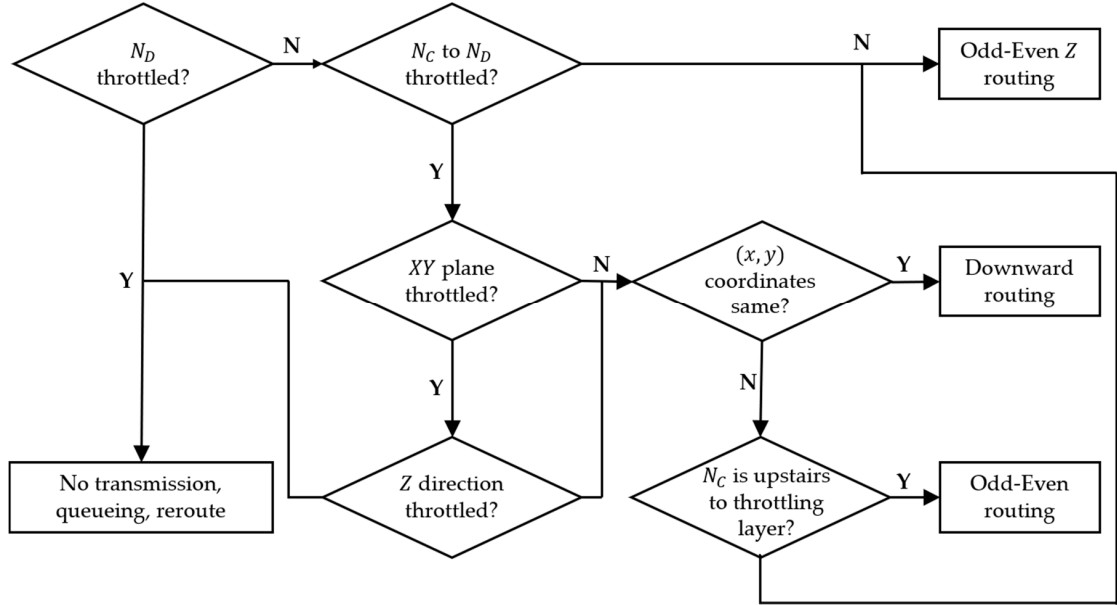

**Figure 3.** Algorithm flow for the routing function of Q-function-based traffic- and thermal-aware adaptive routing (QTTAR).

Figure 4 shows the detailed operations of the routing function for collecting deadlock-free directions in QTTAR, where $N_P$ and $N_n$ are the previous node and a node *n*-hops away from the current node, respectively. Depending on the presence of throttling nodes in the minimal region, the directions are classified into inter-layer and intra-layer. If no throttled node exists in the minimal path region, the routing directions are initially selected through odd–even routing to ensure diversity for the subsequent Q-table-based routing decision phase, as shown in Figure 4a. If a throttling node is detected in the minimal region, intra-layer routing is performed according to the relationship of the x and y coordinates of the nodes $N_C$ and $N_D$. If the x and y coordinate pairs of the two nodes are the same, downward routing occurs in the inter-layer and, if they are different, odd–even routing occurs in the same horizontal plane, as shown in Figure 4b. Through this process, QTTAR achieves reactive RTM by determining the final direction by bypassing the area containing the throttling nodes.

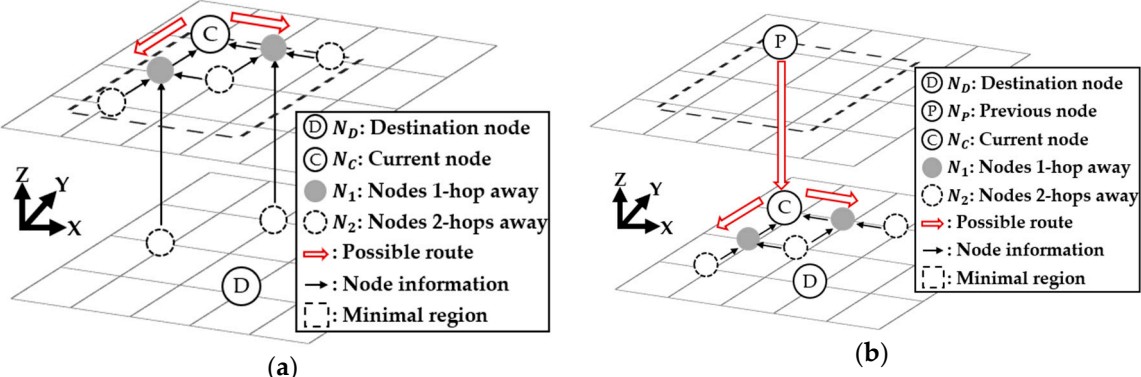

(**a**)                                                                 (**b**)

**Figure 4.** Detailed routability checking method maintaining runtime thermal management (RTM): (**a**) no throttled nodes detected in the minimal region; (**b**) throttled nodes detected.

QTTAR prohibits routing of {North, South, East, West} followed by {Up} to prevent cyclic dependency between inter- and intra-layers, as shown in Figure 5. The {Up} direction routing is performed only when the x and y coordinate pairs of $N_C$ and $N_D$ are all equal and the destination is in the upper layer. This ensures that the directions selected in the first step of QTTAR are deadlock-free.

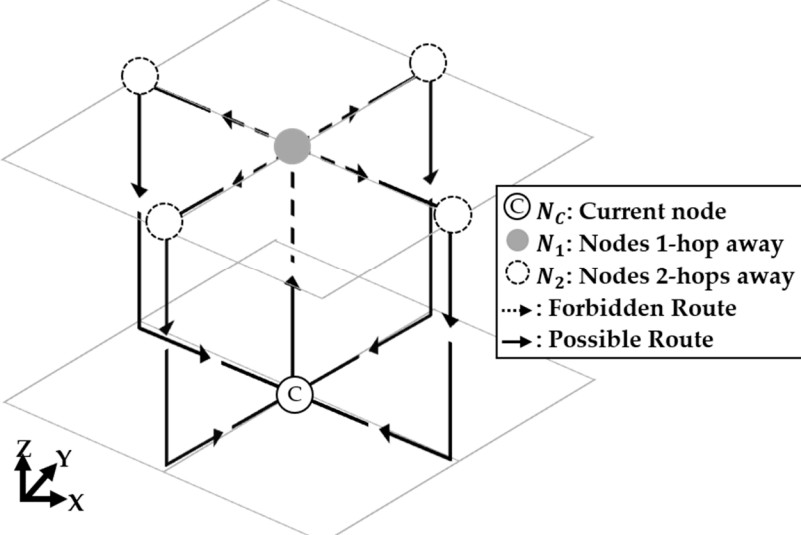

**Figure 5.** Deadlock prevention in the routable direction selection.

### 3.2. Q-Function-Based Routable Direction Selection Strategy (Step 2)

The second step of QTTAR is updating the Q-table based on network congestion information and selecting the direction with the most positive Q-value, as shown in Figure 6. The goal of this step is to select a final direction towards the non-congested region. QTTAR uses Q-learning to estimate the congestion of the network as close to the actual state as possible. Based on the estimated value, a Q-function-based routable direction selection strategy is established. Previous studies that introduced Q-tables in a 2D NoC [21–23] stored estimates for the paths for all nodes in the system in each node. Figure 7a shows a conventional Q-table for a node in a 3D mesh network with an $8 \times 8 \times 4$ topology. In the Q-table, each row corresponds to a destination. Because a separate row is dedicated to each destination in the network and the number of nodes increases in multiples of the number of layers (compared to a 2D NoC), the area overhead of the Q-tables becomes problematic in a 3D NoC. To minimize the size of the Q-table, we simplified the row index to four directions of the intra-layer rather than using the number of destination nodes, as depicted in Figure 7b. In an $n \times m \times l$ 3D mesh, the row size of the Q-table decreases $n \times m \times l$ to four.

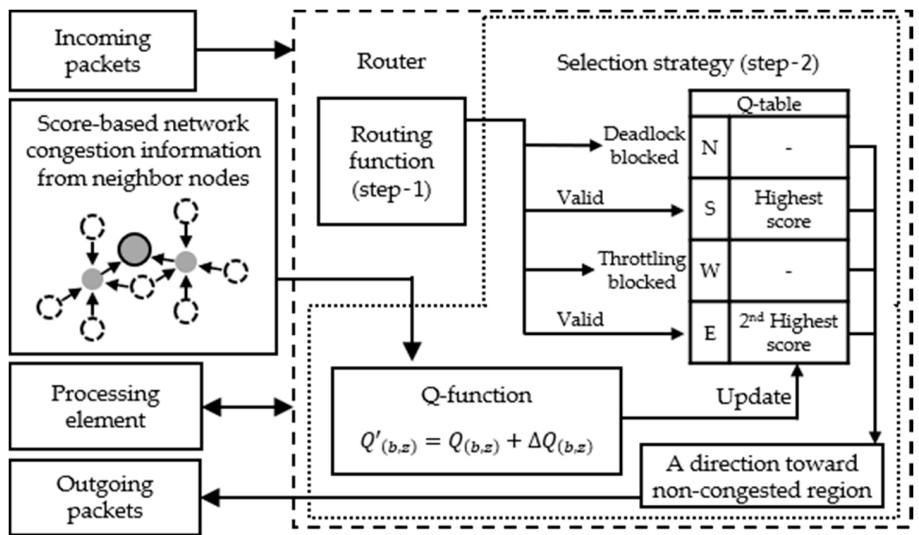

**Figure 6.** Flow of Q-function-based routable direction selection strategy.

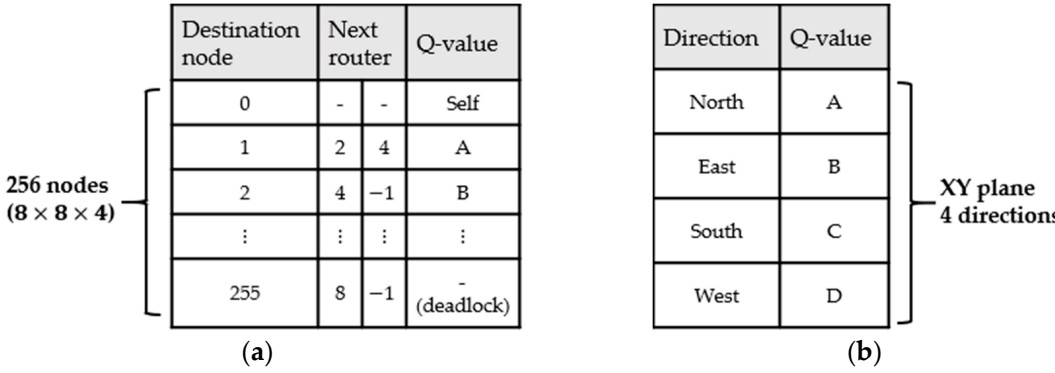

**Figure 7.** Q-table configuration for: (**a**) a conventional Q-table; (**b**) a simplified 2-hop Q-table.

Updating the Q-value in the simplified Q-table is based on information from nodes that are two hops away, because the computational cost of searching all nodes is very high. $N_1$ is one hop away from $N_C$, $N_2$ is two hops away and these nodes belong to the minimal path region in the {North, South, East, West, Up, Down} directions of $N_C$. $N_1$ checks the throttling and buffer queue information of $N_2$ and delivers an estimate to $N_C$. The estimate is a medium for identifying the traffic congestion and throttling state for each direction in $N_C$; a greater value indicates a more positive routing for that direction. Assuming an $8 \times 8 \times 4$ topology, an estimate passed from $N_1$ to $N_C$ can reflect information from up to nine nodes, including one node up and one node down for the minimal region and two hops apart on the same plane, as shown in Figure 8. $N_C$ can identify routing information by considering regional information, rather than network information limited to minimal regions.

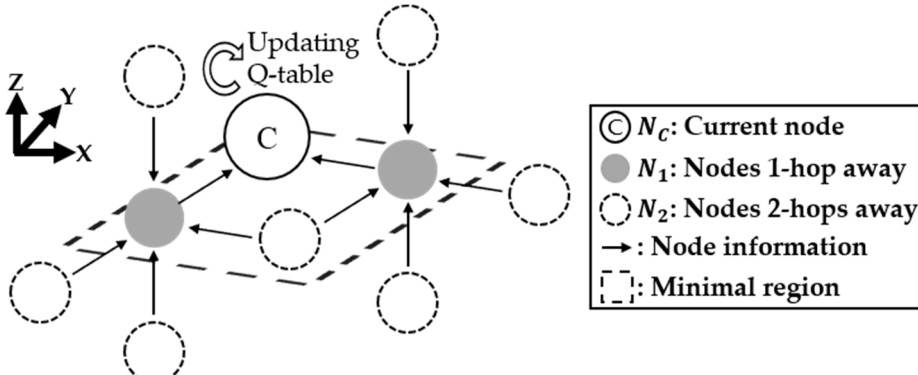

**Figure 8.** Detailed information collection method toward minimal region.

After receiving the estimate for each direction, $N_C$ calculates the new Q-value for the final routing decision, by combining the old Q-value and the estimate. This is expressed in Equation (1) as

$$Q'_{(b,z)} = Q_{(b,z)} + \Delta Q_{(b,z)}, \ z \in \{north, south, east, west\}, \tag{1}$$

where $Q_{(b,z)}$ represents the old Q-value being updated from the past to the present when routing from the current node to direction $z$ and $Q'_{(b,z)}$ is the new Q-value updated for the packet to be routed, calculated as the sum of $Q_{(b,z)}$ and $\Delta Q_{(b,z)}$. In this case, the larger the Q-value, the more positive the value of $Q'_{(b,z)}$. The parameter $\Delta Q_{(b,z)}$ is the delta estimate for modification and is expressed in Equation (2) as

$$\Delta Q_{(b,z)} = \alpha \cdot \left( \sum S_{(neighbor,y)} - Q_{(b,z)} \right), \ y \in \{north, south, east, west, up, down\}, \tag{2}$$

where $S_{(neighbor,y)}$ represents the sum of newly received estimates from neighbor nodes in each direction, i.e., the scores implying input buffer state of neighbor nodes. $\Delta Q_{(b,z)}$ act as modifiers to update the new Q-values using old Q-values and estimates from adjacent nodes. Algorithm 1 presents the mechanism for updating the Q-table in each node.

The parameter $\alpha$ is a learning rate for determining the weight of a delta estimate. The learning rate is a value between zero and one and determines how much to overwrite the old value by when reflecting new information to the Q-table. We can observe the best average latency when the learning rate is 0.6 (as determined through empirical experiments) and we used a learning rate of 0.6 in all experiments. Figure 9 shows the results from analyzing a learning rate of 0.6 compared to other values. A learning rate of 0.3 indicates that the old data have a heavy weight and 0.9 indicates that the new data have a heavy weight. Through this process, each node in the network updates the Q-table with the newly calculated $Q'_{(b,z)}$, selects the direction with the largest value and then performs routing.

---

**Algorithm 1** Updating Q-table.

---

1:     **Set** *All directions ← {north, south, east, west, up, down}*
2:     **Set** *Neighbor data[i] ← {validation, number of available buffer slot, throttling} from direction i*
3:     **Set** *Port data[i] ← Neighbor data [All directions] received from i-th port*
4:     **Set** *Q-value[i] ← Estimates for direction i*
5:
6:     **Variables**
7:        *Bool validation*
8:        *Bool throttling*
9:        *Integer free_slots*
10:       *Integer Temporal_score*
11:
12:    **Function** Update
13:    **for** *i* = 0 to size (*All directions*) **do**
14:       *Temporal data ← Port data[i]*
15:       **for** *j* = 0 to size (*All directions*) **do**
16:          **if** *Temporal data.Neighbor data[j].validation* == valid
17:             *validation* ← true
18:          **else** *validation* ← false
19:
20:          **if** *Temporal data.Neighbor data[j].throttling* == throttled
21:             *throttling* ← true
22:          **else** *throttling* ← false
23:
24:          *free_slots ← Temporal data.Neighbor data[j].buffer slot*
25:          *Temporal_score* += (*int*) *validation × throttling × free_slots*
26:       **end for**
27:
28:       *Q-value[i]* ← (1 - Learning_rate) * *Q_value[i]* + Learning_rate * *Temporal_score*
29:    **end for**
30:    **end function**

---

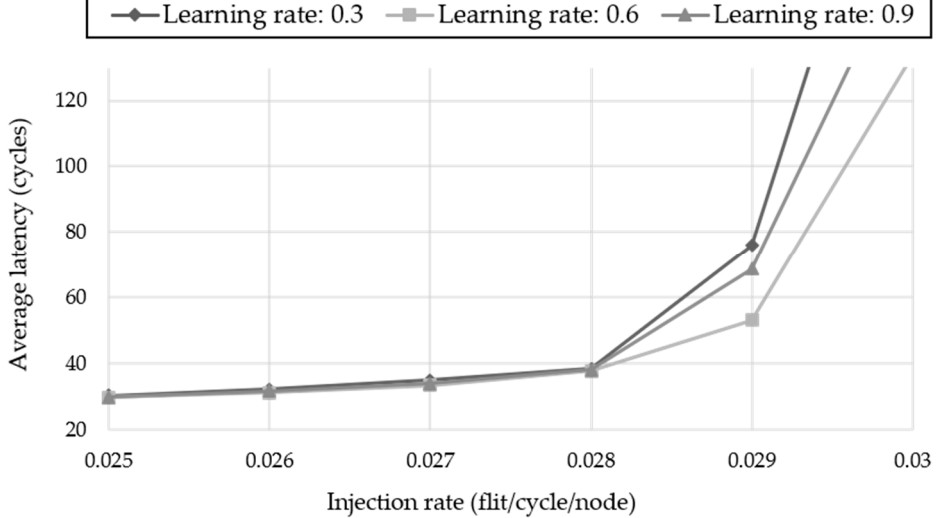

**Figure 9.** Analysis of the best performance-yielding learning rate.

To minimize hardware complexity from floating-point multiplication, which is a drawback of Q-learning-based routing, we devised a method to reduce the amount of computation. QTTAR reads a pre-calculated value from the look-up table (LUT) inside the router, rather than performing multiplication operations every time. Based on the architecture information, the range of calculated values is pre-determined and divided into four sections. If the estimates belong to one section, the method reads the contents of the LUT, as shown in Table 2. This method is used to calculate $\Delta Q_{(b,z)}$, which changes the multiplication process to a simple index and reduces the area overhead of the floating-point multipliers. $S_{max}$ is the maximum value among $\sum S_{(neighbor,y)}$ and is expressed as the product of six directions {North, South, East, West, Up, Down} and the buffer flit size. $S_{max}$ is expressed in Equation (3) as

$$S_{max} = max\left\{\sum S_{(neighbor,y)}\right\} = \text{buffer flit size} \times 6, \tag{3}$$

**Table 2.** Look-up table for simplified multiplication.

| Range | Value |
|-------|-------|
| $0 \leq \sum S_{(neighbor,y)} < 0.2 \times S_{max}$ | $0.1 \times S_{max}$ |
| $0.2 \times S_{max} \leq \sum S_{(neighbor,y)} < 0.5 \times S_{max}$ | $0.35 \times S_{max}$ |
| $0.5 \times S_{max} \leq \sum S_{(neighbor,y)} < 0.8 \times S_{max}$ | $0.65 \times S_{max}$ |
| $0.8 \times S_{max} \leq \sum S_{(neighbor,y)} < S_{max}$ | $0.9 \times S_{max}$ |

## 4. Simulation Results

AccessNoxim [27], which provides the co-simulation and 3D configuration, was used to demonstrate the advantage of the proposed method. For the comparison of various routing algorithms, an NoC simulator, Noxim [28] and an architecture-level thermal model, HotSpot [29], were applied together. Table 3 presents the simulation parameters used for co-simulation comprising the network, power and thermal models of a 3D NoC, which are default values in AccessNoxim [27]. Each 3D router contains two extra physical channels {Up, Down} for vertical connections and the depth of each input buffer is 16 flits without a virtual channel. We evaluated network performance, including average latency, throughput, traffic load and temperature distribution with two related studies. One was the topology-aware routing proposed in [17] (referred to as 'TAAR') and the other employed the traffic- and thermal-aware routing method proposed in [6] (referred to as 'TTABR').

**Table 3.** Specification of parameters for simulation [27].

| Parameter | Value |
|-----------|-------|
| Packet size | 8 flits |
| Buffer size | 16 flits |
| Simulation time | $5 \times 10^5$ cycles |
| Warm-up time | 4000 cycles |
| Mesh size | $8 \times 8 \times 4$ |
| Throttling scheme | vertical throttling |
| Traffic pattern | random, transpose-1, shuffle, bit-reversal |
| Temperature threshold | 98 °C |
| Initial temperature | 80 °C |

### 4.1. Performance

Figure 10 shows an average latency comparison for random, transpose-1, shuffle and bit-reversal traffic patterns for the same simulation time. The corresponding throughput comparison is shown in Figure 11. QTTAR showed performance improvement over TAAR and TTABR, regardless of the traffic pattern. Although TTABR provides greater path diversity than TAAR, the similar routing function produces almost identical latency. QTTAR achieved the highest latency improvement (3.98 times)

in the transpose-1 traffic pattern, as shown in Figure 10b. Considering that the transpose-1 traffic pattern focuses on burst communication to a specific node, the result indicates that QTTAR effectively distributes packets that have the same destination. Thus, the proposed algorithm reflects the regional realistic congestion state in more detail through the Q-function, based on improved routing diversity and improved congestion-aware routing compared to other algorithms. Additionally, QTTAR exhibited a throughput improvement in the range of 14.0%–28.2% compared with TAAR and TTABR.

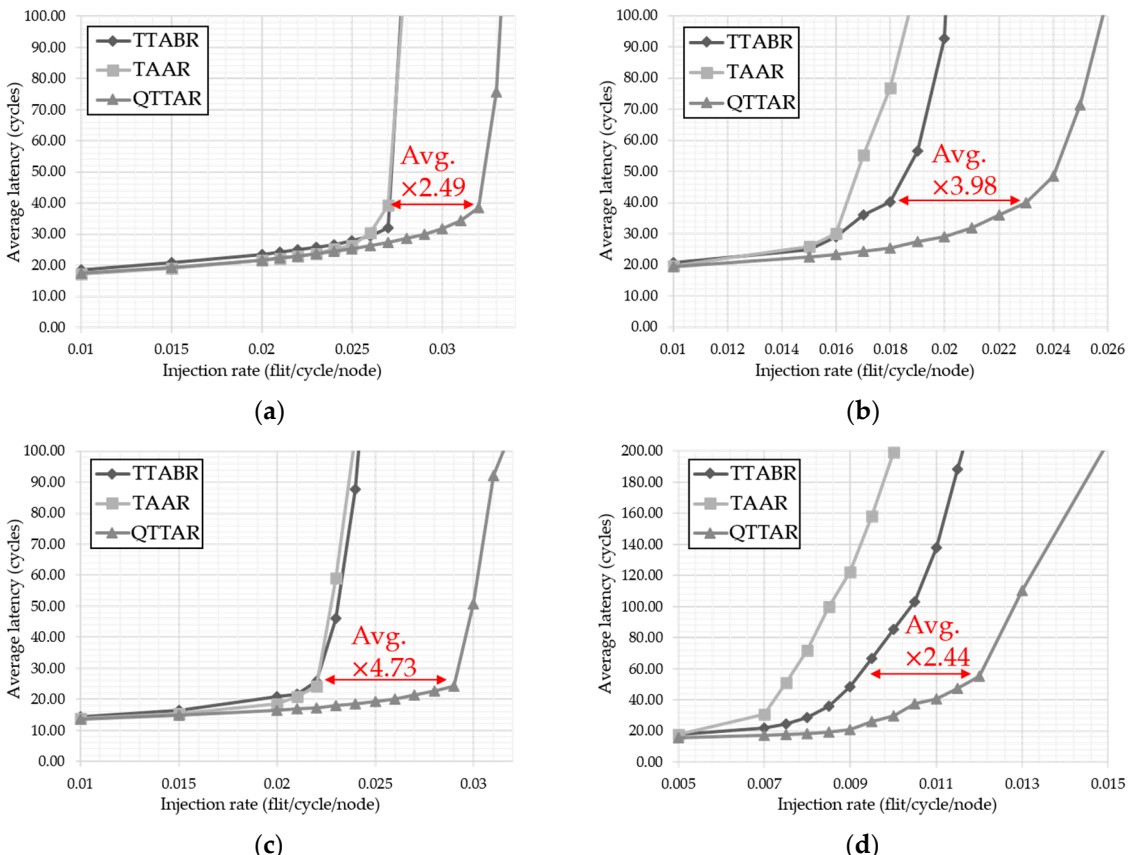

**Figure 10.** Latency comparison under different traffic conditions: (**a**) random; (**b**) transpose-1; (**c**) shuffle; (**d**) bit-reversal.

## 4.2. Traffic Load Distributions

Figure 12 shows the traffic load distributions under random, transpose-1, shuffle and bit-reversal traffic patterns. The averages and standard deviations for the traffic load distributions are presented in Table 4. QTTAR exhibited 24.9% more balanced inter-layer traffic load on average compared to TAAR and TTABR. Because TTABR and TAAR employed a similar routing function, their traffic load distributions are similar. The standard deviation of each node in the QTTAR is not significantly different from that of the other algorithms, but the standard deviation of the inter-layer traffic load distribution is the lowest in all but the shuffle traffic pattern. This is because TAAR and TTABR perform inter-layer routing only after confirming the intra-layer adaptability, without considering the inter-layer traffic load information. Therefore, if the destination node is located in a different layer, downward routing prevails, causing an unbalanced inter-layer traffic load distribution. Considering that the shuffle traffic pattern is mainly for intra-layer communication, the scale of packets processed per layer significantly influences the inter-layer traffic load distribution. Consequently, the inter-layer traffic load distribution of the QTTAR is unbalanced only in the shuffle traffic pattern.

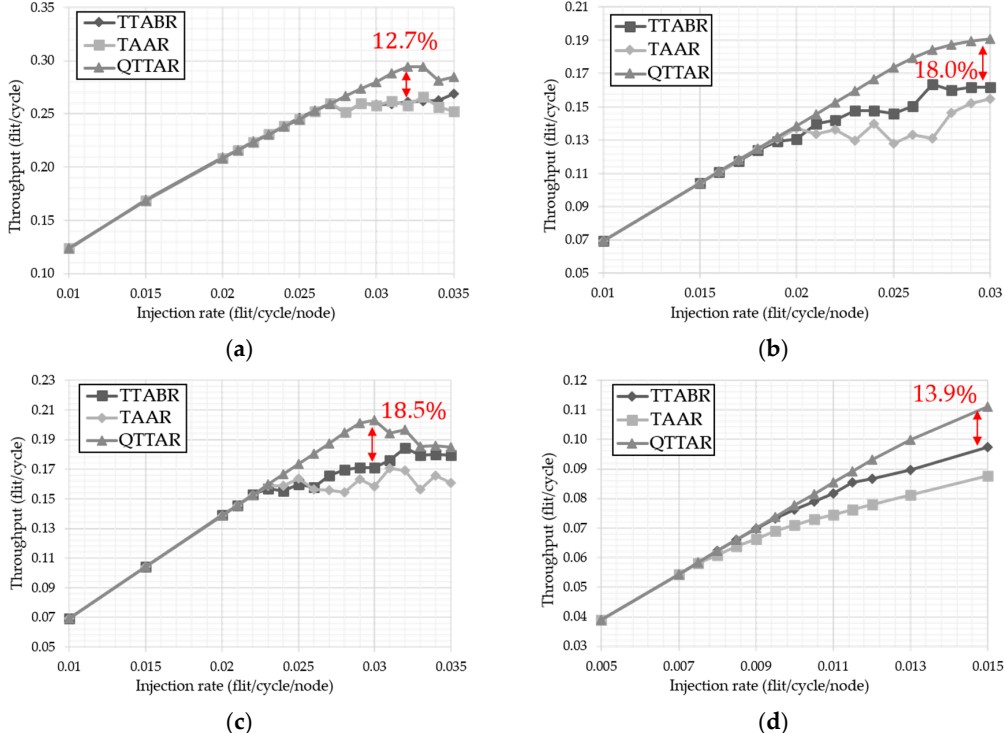

**Figure 11.** Throughput comparison under different traffic conditions: (**a**) random; (**b**) transpose-1; (**c**) shuffle; (**d**) bit-reversal.

**Table 4.** Traffic load and temperature distribution comparison under different conditions for thermal-aware adaptive beltway routing (TTABR), topology-aware adaptive routing (TAAR) and QTTAR.

| | Random | | | | | |
|---|---|---|---|---|---|---|
| | **Traffic load distribution** | | | **Temperature distribution** | | |
| Algorithms | TTABR | TAAR | QTTAR | TTABR | TAAR | QTTAR |
| Avg. | 678,252 | 646,493 | 683,231 | 93.41 | 92.92 | 93.19 |
| Stdv. | 172,380 | 216,223 | 217,058 | 3.10 | 3.39 | 3.26 |
| Inter-layer Stdv. | 108,680 | 92,388 | 88,832 | 0.98 | 0.83 | 0.36 |
| | **Transpose-1** | | | | | |
| | **Traffic load distribution** | | | **Temperature distribution** | | |
| Algorithms | TTABR | TAAR | QTTAR | TTABR | TAAR | QTTAR |
| Avg. | 544,676 | 608,262 | 609,948 | 90.86 | 91.46 | 91.99 |
| Stdv. | 219,945 | 216,159 | 201,076 | 2.90 | 3.23 | 3.25 |
| Inter-layer Stdv. | 143,969 | 124,835 | 96,728 | 1.01 | 0.82 | 0.24 |
| | **Shuffle** | | | | | |
| | **Traffic load distribution** | | | **Temperature distribution** | | |
| Algorithms | TTABR | TAAR | QTTAR | TTABR | TAAR | QTTAR |
| Avg. | 461,710 | 488,110 | 563,517 | 90.66 | 90.91 | 91.80 |
| Stdv. | 171,942 | 200,824 | 179,000 | 2.86 | 3.09 | 3.22 |
| Inter-layer Stdv. | 38,188 | 49,838 | 45,813 | 0.35 | 0.30 | 0.38 |
| | **Bit-reversal** | | | | | |
| | **Traffic load distribution** | | | **Temperature distribution** | | |
| Algorithms | TTABR | TAAR | QTTAR | TTABR | TAAR | QTTAR |
| Avg. | 285,068 | 249,169 | 276,730 | 88.40 | 87.92 | 88.25 |
| Stdv. | 159,237 | 174,110 | 144,133 | 1.98 | 2.02 | 2.07 |
| Inter-layer Stdv. | 45,284 | 40,791 | 14,320 | 0.28 | 0.22 | 0.23 |

*Unit : flit (traffic load distribution), °C (temperatrue distribution).*

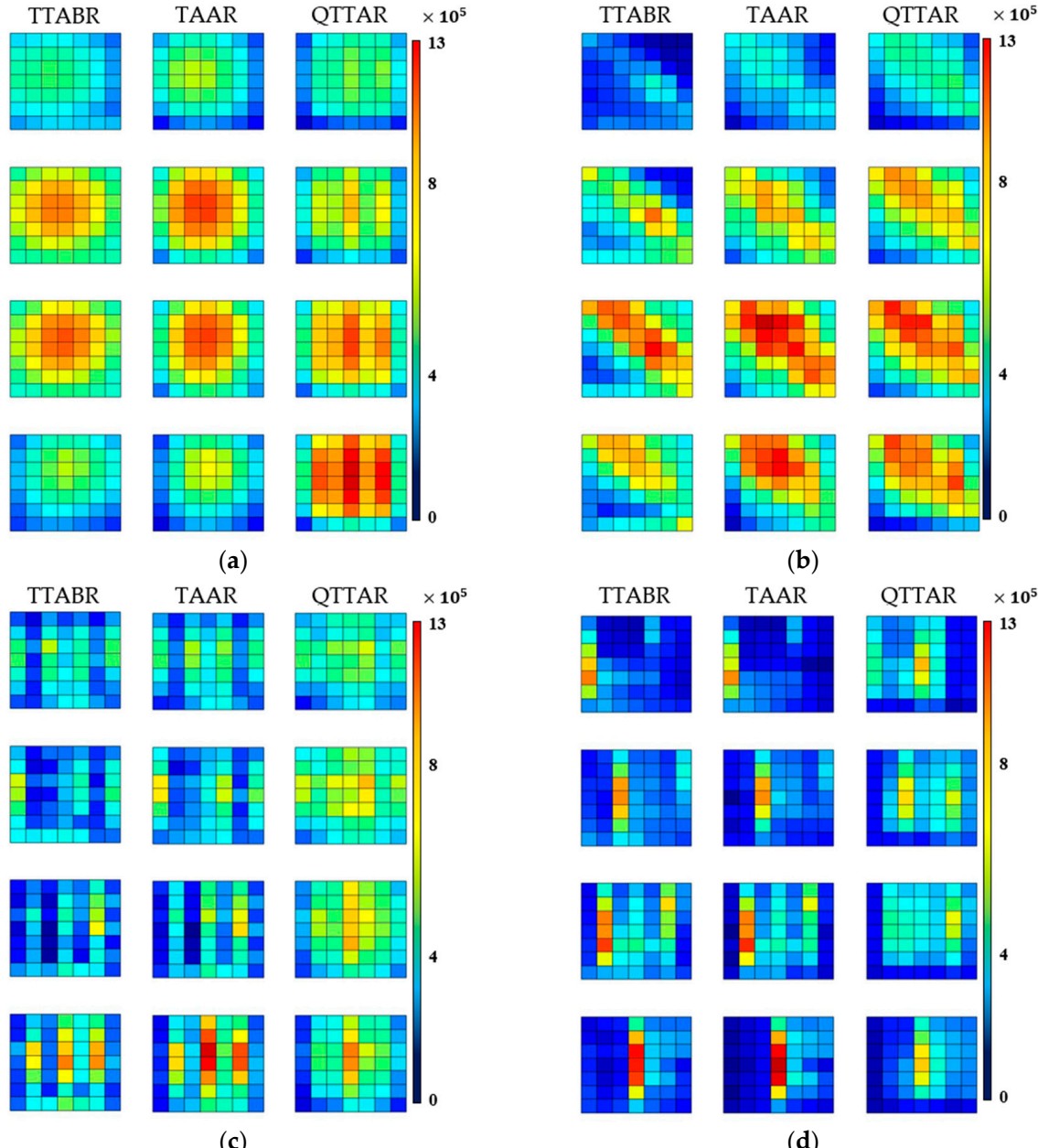

**Figure 12.** Traffic load distribution comparison under different traffic conditions: (**a**) random; (**b**) transpose-1; (**c**) shuffle; (**d**) bit-reversal.

### 4.3. Temperature Distributions for Traffic Patterns

Figure 13 illustrates the temperature distributions for random, transpose-1, shuffle and bit-reversal traffic patterns. The averages and standard deviations of the temperature distributions are presented in Table 4. Because the QTTAR improved the throughput by 14.0%–28.2%, the temperature and standard deviation increased by 1.3% and 4.7% on average, respectively. However, the inter-layer standard deviation for temperature was reduced by 30.6% on average, thereby indicating a more balanced temperature distribution by layer. This is because the QTTAR sent packets in a direction that avoided the high-temperature regions associated with excessive congestion, similar to the traffic load distribution by layer shown in Figure 12.

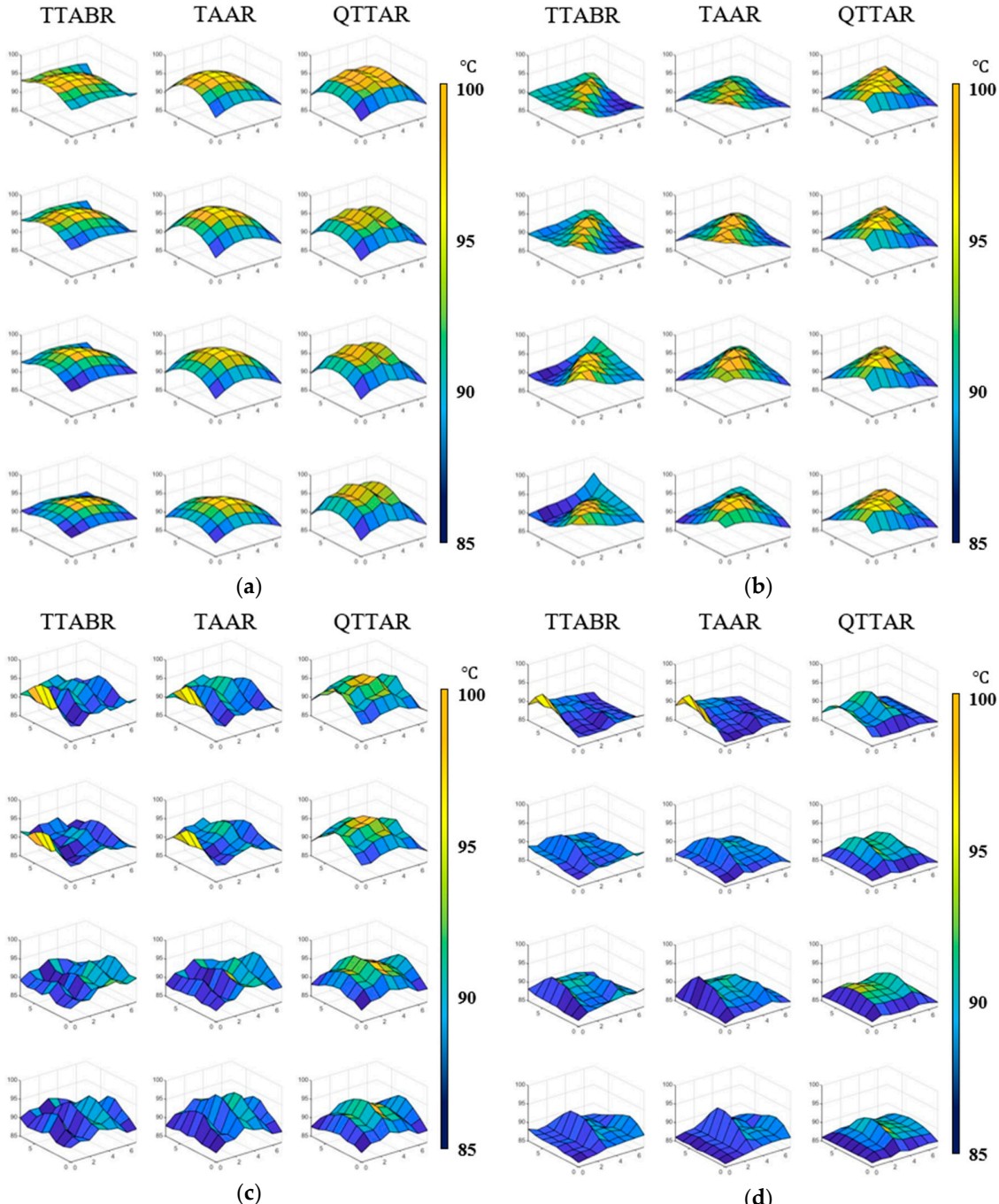

**Figure 13.** Temperature distributions for different traffic conditions: (**a**) random; (**b**) transpose-1; (**c**) shuffle; (**d**) bit-reversal.

## 5. Conclusions

Three-dimensional die-stacked NoCs have a wider bandwidth and lower packet latency than 2D NoCs, which have significant drawbacks, such as increased thermal density and unbalanced heat dissipation. Various RTM techniques have been introduced to reduce hotspots. However, inter-layer traffic imbalances and performance degradations remain. In this study, we proposed a Q-function-based 3D NoC routing algorithm with a learning scheme, to mitigate the performance degradations of existing 3D NoC routing algorithms based on RTM. QTTAR gradually learns the network conditions and determines the optimal paths for each direction in the runtime, establishing a routing policy that considers the congested region more accurately, while maintaining the RTM. The

simulation results revealed that QTTAR reduced latency and improved throughput by 14.0%–28.2% compared to previous studies, due to its more balanced inter-layer traffic loads. Furthermore, QTTAR provided effective inter-layer thermal management, reducing the standard deviation of the inter-layer temperature distribution by 30.6% on average.

**Author Contributions:** Conceptualization, S.C.L. and T.H.H.; methodology, S.C.L.; software, S.C.L.; validation, S.C.L. and T.H.H.; formal analysis, T.H.H.; investigation, S.C.L.; resources, T.H.H.; data curation, S.C.L.; writing—original draft preparation, S.C.L.; writing—review and editing, T.H.H.; visualization, S.C.L.; supervision, T.H.H.; project administration, T.H.H.; funding acquisition, T.H.H. All authors have read and agreed to the published version of the manuscript.

**Funding:** This work was supported, in part, by the Ministry of Trade, Industry and Energy (MOTIE) and Korea Semiconductor Research Consortium (KSRC) support program (10080594) for the development of the future semiconductor device and, in part, by the Institute of Information and communications Technology Planning and Evaluation (IITP) grant, funded by the Korean government (MSIT) (No.2019-0-00421, AI Graduate School Support Program (Sungkyunkwan University)).

**Conflicts of Interest:** The authors declare no conflict of interest.

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
