# Peer review of "Q-Function-Based Traffic- and Thermal-Aware Adaptive Routing for 3D Network-on-Chip"

_electronics, doi:10.3390/electronics9030392_

Round 1

Reviewer 1 Report

In this article, the authors propose a Q-function-based 3D NoC routing algorithm with a learning scheme, to mitigate the performance degradations of existing 3D NoC routing algorithms based on RTM. QTTAR gradually learns the network conditions and determines the optimal paths for each direction in run-time, establishing a routing policy that considers the congested region more accurately while maintaining the RTM. The simulation results have revealed that QTTAR reduced latency and improved throughput by 0.14-0.28 compared to previous studies, due to its more balanced inter-layer traffic loads. Furthermore, QTTAR has provided effective inter-layer thermal management, reducing the standard deviation of the inter-layer temperature distribution by 0.31 on average.

The paper is well written and motivated. According to the simulation results, It has enough novel and significant ingredients to be published as a journal paper. The quality of the presentation is good, and overall the research is interesting for researchers working on this domain. Thus, I would like to recommend this paper for its publication in this journal.

Some issues that the authors should address:

  1. All figures before the simulation section are from other publications? If that is the case, there is no reference to the original source. 
  2. Literature survey section includes the proposed work part. The literature survey is good. However, proposed work is given only in the last paragraph (less content for proposed work).
  3. If the proposed algorithm can be added, it will improve the content.
  4. Are the simulation parameters directly from the AccessNoxim simulator or the authors have made any changes in the values? If so, specify the reason. If not, reference is missing in the parameter table.
  5. What changes have been done in the back end code for implementation (proposed work), detail should be there. 

The paper looks good for acceptance. 

Author Response

Dear Editor and Reviewers,

We greatly appreciate your efforts and valuable comments on our manuscript. All comments by the reviewer have been addressed, with corresponding changes made directly to the manuscript where appropriate.

Reviewer 2 Report

The problem is timely and the used strategy is smart. The algorithm that manage heat distributions are welcome in innovative device both as regards computational units and other class of systems. The management of heat distribution arise also in micro-nanofluidic devices used for more applications like systems on chip, fluid network on chip and so on. The idea developed in the paper is in some sense in progress therefore in other systems. I suggest the authors to include the following paper:

Energies Open AccessVolume 12, Issue 13, 2019, Article number 2556

A real time feed forward control of slug flow in microchannels(Article)(Open Access)

  • Gagliano, S.,

  • Cairone, F.,

  • Amenta, A.,

They could further stress the concept that their approach is quite general

and timely for various applications.

Author Response

(The authors gave the same response as above.)
